# Dynamically Unfolding Recurrent Restorer: A Moving Endpoint Control Method for Image Restoration

**Xiaoshuai Zhang**[*]
Institute of Computer Science and Technology,
Peking University
jet@pku.edu.cn

**Yiping Lu**[*]
School Of Mathmatical Science,
Peking university
luyiping9712@pku.edu.cn

**Jiaying Liu**
Institute of Computer Science and Technology,
Peking University
liujiaying@pku.edu.cn

**Bin Dong**
Beijing International Center for Mathematical Research, Peking University
Center for Data Science, Peking University
Beijing Institute of Big Data Research,
Beijing, China
dongbin@math.pku.edu.cn

## ABSTRACT

In this paper, we propose a new control framework called the moving endpoint control to restore images corrupted by different degradation levels using a single model. The proposed control problem contains an image restoration dynamic which is modeled by a convolutional RNN. The moving endpoint, which is essentially the terminal time of the associated dynamic, is determined by a policy network. We call the proposed model the dynamically unfolding recurrent restorer (DURR). Numerical experiments show that DURR is able to achieve state-of-the-art performances on blind image denoising and JPEG image deblocking. Furthermore, DURR can well generalize to images with higher degradation levels that are not included in the training stage.[1]

## 1 INTRODUCTION

Image restoration, including image denoising, deblurring, inpainting, *etc.*, is one of the most important areas in imaging science. Its major purpose is to obtain high quality reconstructions of images corrupted in various ways during imaging, acquisiting, and storing, and enable us to see crucial but subtle objects that reside in the images. Image restoration has been an active research area. Numerous models and algorithms have been developed for the past few decades. Before the uprise of deep learning methods, there were two classes of image restoration approaches that were widely adopted in the field: transformation based approach and PDE approach. The transformation based approach includes wavelet and wavelet frame based methods (Elad et al., 2005; Starck et al., 2005; Daubechies et al., 2007; Cai et al., 2009), dictionary learning based methods (Aharon et al., 2006), similarity based methods (Buades et al., 2005; Dabov et al., 2007), low-rank models (Ji et al., 2010; Gu et al., 2014), *etc.* The PDE approach includes variational models (Mumford & Shah, 1989; Rudin et al., 1992; Bredies et al., 2010), nonlinear diffusions (Perona & Malik, 1990; Catté et al., 1992; Weickert, 1998), nonlinear hyperbolic equations (Osher & Rudin, 1990), *etc.* More recently, deep connections

---

[*]Equal contribution.
[1]Supplementary materials and full code can be found at the project page.

between wavelet frame based methods and PDE approach were established (Cai et al., 2012; 2016; Dong et al., 2017).

One of the greatest challenge for image restoration is to properly handle image degradations of different levels. In the existing transformation based or PDE based methods, there is always at least one tuning parameter (*e.g.* the regularization parameter for variational models and terminal time for nonlinear diffusions) that needs to be manually selected. The choice of the parameter heavily relies on the degradation level.

Recent years, deep learning models for image restoration tasks have significantly advanced the state-of-the-art of the field. Jain & Seung (2009) proposed a convolutional neural network (CNN) for image denoising which has better expressive power than the MRF models by Lan et al. (2006). Inspired by nonlinear diffusions, Chen & Pock (2017) designed a deep neural network for image denoising and Zhang et al. (2017a) improves the capacity by introducing a deeper neural network with residual connections. Chen et al. (2017) use the CNN to simulate a wide variety of image processing operators, achieving high efficiencies with little accuracy drop. However, these models cannot gracefully handle images with varied degradation levels. Although one may train different models for images with different levels, this may limit the application of these models in practice due to lack of flexibility.

Taking blind image denoising for example. Zhang et al. (2017a) designed a 20-layer neural network for the task, called DnCNN-B, which had a huge number of parameters. To reduce number of parameters, Lefkimmiatis (2017) proposed the UNLNet$_5$, by unrolling a projection gradient algorithm for a constrained optimization model. However, Lefkimmiatis (2017) also observed a drop in PSNR comparing to DnCNN. Therefore, the design of a light-weighted and yet effective model for blind image denoising remains a challenge. Moreover, deep learning based models trained on simulated gaussian noise images usually fail to handle real world noise, as will be illustrated in later sections.

Another example is JPEG image deblocking. JPEG is the most commonly used lossy image compression method. However, this method tend to introduce undesired artifacts as the compression rate increases. JPEG image deblocking aims to eliminate the artifacts and improve the image quality. Recently, deep learning based methods were proposed for JPEG deblocking (Dong et al., 2015; Zhang et al., 2017a; 2018). However, most of their models are trained and evaluated on a given quality factor. Thus it would be hard for these methods to apply to Internet images, where the quality factors are usually unknown.

In this paper, we propose a single image restoration model that can robustly restore images with varied degradation levels even when the degradation level is well outside of that of the training set. Our proposed model for image restoration is inspired by the recent development on the relation between deep learning and optimal control. The relation between supervised deep learning methods and optimal control has been discovered and exploited by Weinan (2017); Lu et al. (2018); Chang et al. (2017); Fang et al. (2017). The key idea is to consider the residual block $x_{n+1} = x_n + f(x_n)$ as an approximation to the continuous dynamics $\dot{X} = f(X)$. In particular, Lu et al. (2018); Fang et al. (2017) demonstrated that the training process of a class of deep models (*e.g.* ResNet by He et al. (2016), PolyNet by Zhang et al. (2017b), *etc.*) can be understood as solving the following control problem:

$$\min_w \left( L(X(T), y) + \int_0^\tau R(w(t), t) dt \right)$$
$$s.t. \ \dot{X} = f(X(t), w(t)), t \in (0, \tau) \tag{1}$$
$$X(0) = x_0.$$

Here $x_0$ is the input, $y$ is the regression target or label, $\dot{X} = f(X, w)$ is the deep neural network with parameter $w(t)$, $R$ is the regularization term and $L$ can be any loss function to measure the difference between the reconstructed images and the ground truths.

In the context of image restoration, the control dynamic $\dot{X} = f(X(t), \omega(t)), t \in (0, \tau)$ can be, for example, a diffusion process learned using a deep neural network. The terminal time $\tau$ of the diffusion corresponds to the depth of the neural network. Previous works simply fixed the depth of the network, *i.e.* the terminal time, as a fixed hyper-parameter. However Mrázek & Navara (2003) showed that

the optimal terminal time of diffusion differs from image to image. Furthermore, when an image is corrupted by higher noise levels, the optimal terminal time for a typical noise removal diffusion should be greater than when a less noisy image is being processed. This is the main reason why current deep models are not robust enough to handle images with varied noise levels. In this paper, we no longer treat the terminal time as a hyper-parameter. Instead, we design a new architecture (see Fig. 3) that contains both a deep diffusion-like network and another network that determines the optimal terminal time for each input image. We propose a novel moving endpoint control model to train the aforementioned architecture. We call the proposed architecture the dynamically unfolding recurrent restorer (DURR).

We first cast the model in the continuum setting. Let $x_0$ be an observed degraded image and $y$ be its corresponding damage-free counterpart. We want to learn a time-independent dynamic system $\dot{X} = f(X(t), w)$ with parameters $w$ so that $X(0) = x$ and $X(\tau) \approx y$ for some $\tau > 0$. See Fig. 2 for an illustration of our idea. The reason that we do not require $X(\tau) = y$ is to avoid over-fitting. For varied degradation levels and different images, the optimal terminal time $\tau$ of the dynamics may vary. Therefore, we need to include the variable $\tau$ in the learning process as well. The learning of the dynamic system and the terminal time can be gracefully casted as the following moving endpoint control problem:

$$
\begin{aligned}
\min_{w, \tau(x)} \ & L(X(\tau), y) + \int_0^{\tau(x)} R(w(t), t) dt \\
s.t. \ & \dot{X} = f(X(t), w(t)), t \in (0, \tau(x)) \\
& X(0) = x.
\end{aligned} \tag{2}
$$

Different from the previous control problem, in our model the terminal time $\tau$ is also a parameter to be optimized and it depends on the data $x$. The dynamic system $\dot{X} = f(X(t), w)$ is modeled by a recurrent neural network (RNN) with a residual connection, which can be understood as a residual network with shared weights (Liao & Poggio, 2016). We shall refer to this RNN as the *restoration unit*. In order to learn the terminal time of the dynamics, we adopt a *policy network* to adaptively determine an optimal stopping time. Our learning framework is demonstrated in Fig. 3. We note that the above moving endpoint control problem can be regarded as the penalized version of the well-known fixed endpoint control problem in optimal control (Evans, 2005), where instead of penalizing the difference between $X(\tau)$ and $y$, the constraint $X(\tau) = y$ is strictly enforced.

In short, we summarize our contribution as following:

- We are the first to use convolutional RNN for image restoration with unknown degradation levels, where the unfolding time of the RNN is determined dynamically at run-time by a policy unit (could be either handcrafted or RL-based).

- The proposed model achieves state-of-the-art performances with significantly less parameters and better running efficiencies than some of the state-of-the-art models.

- We reveal the relationship between the generalization power and unfolding time of the RNN by extensive experiments. The proposed model, DURR, has strong generalization to images with varied degradation levels and even to the degradation level that is unseen by the model during training (Fig. 1).

- The DURR is able to well handle real image denoising without further modification. Qualitative results have shown that our processed images have better visual quality, especially sharper details compared to others.

## 2 METHOD

The proposed architecture, *i.e.* DURR, contains an RNN (called the restoration unit) imitating a nonlinear diffusion for image restoration, and a deep policy network (policy unit) to determine the terminal time of the RNN. In this section, we discuss the training of the two components based on our moving endpoint control formulation. As will be elaborated, we first train the restoration unit to determine $\omega$, and then train the policy unit to estimate $\tau(x)$.

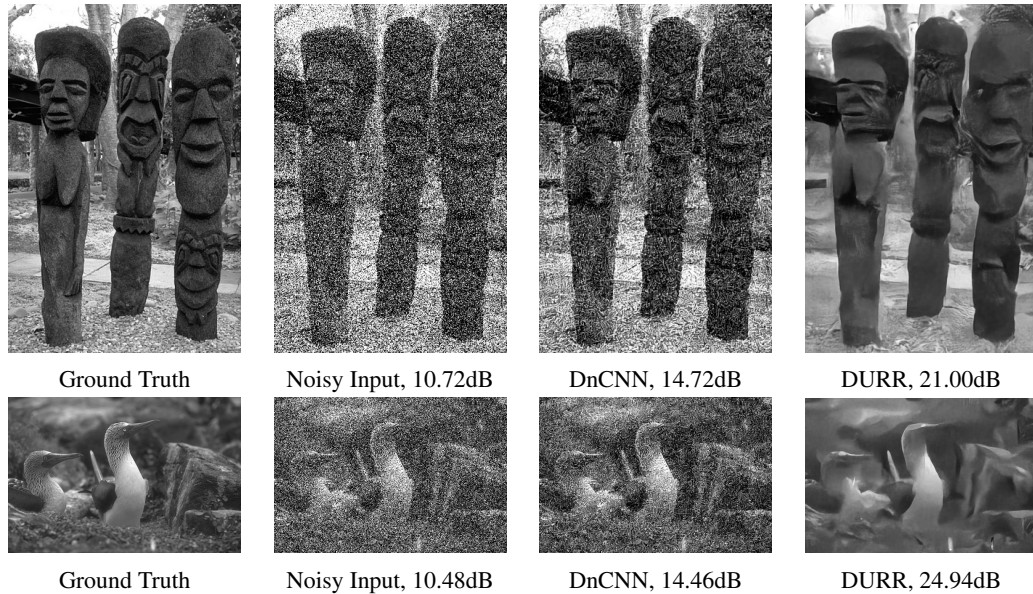

| Ground Truth | Noisy Input, 10.72dB | DnCNN, 14.72dB | DURR, 21.00dB |

| Ground Truth | Noisy Input, 10.48dB | DnCNN, 14.46dB | DURR, 24.94dB |

Figure 1: Denoising results of images from BSD68 under extreme noise conditions not seen in training data ($\sigma = 95$).

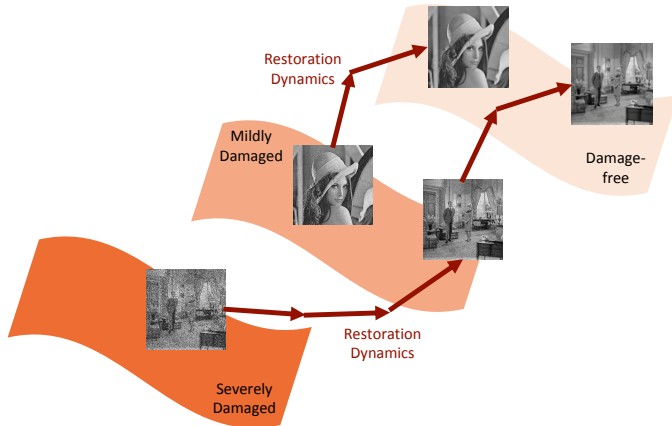

Figure 2: The proposed moving endpoint control model: evolving a learned reconstruction dynamics and ending at high-quality images.

## 2.1 TRAINING THE RESTORATION UNIT

If the terminal time $\tau$ for every input $x_i$ is given (*i.e.* given a certain policy), the restoration unit can be optimized accordingly. We would like to show in this section that the policy used during training greatly influences the performance and the generalization ability of the restoration unit. More specifically, a restoration unit can be better trained by a good policy.

The simplest policy is to fix the loop time $\tau$ as a constant for every input. We name such policy as "naive policy". A more reasonable policy is to manually assign an unfolding time for each degradation level during training. We shall call this policy the "refined policy". Since we have not trained the policy unit yet, to evaluate the performance of the trained restoration units, we manually pick the output image with the highest PSNR (*i.e.* the peak PSNR).

We take denoising as an example here. The peak PSNRs of the restoration unit trained with different policies are listed in Table. 1. Fig. 4 illustrates the average loop times when the peak PSNRs appear. The training is done on both single noise level ($\sigma = 40$) and multiple noise levels ($\sigma = 35, 45$). For

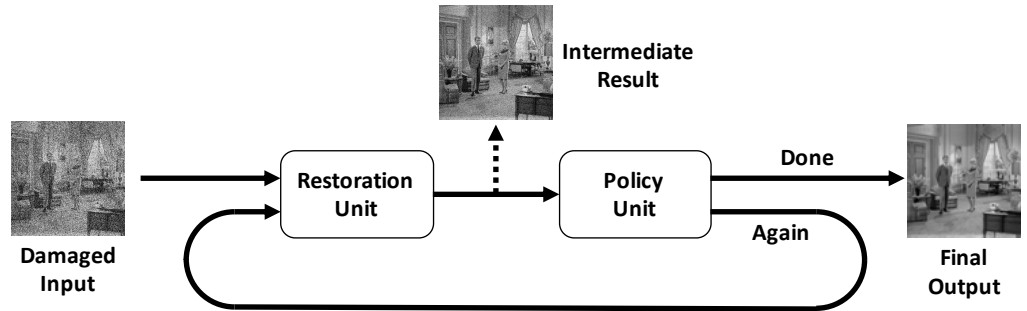

Figure 3: Pipeline of the dynamically unfolding recurrent restorer (DURR).

the refined policy, the noise levels and the associated loop times are (35, 6), (45, 9). For the naive policy, we always fix the loop times to 8.

Table 1: Average peak PSNR on BSD68 with different training strategies.

| Strategy | | Noise Level | | | | | | |
|---|---|---|---|---|---|---|---|---|
| Training Noise | Policy | 25 | 30 | 35 | 40 | 45 | 50 | 55 |
| 40 | Naive | 28.61 | 28.13 | 27.62 | **27.19** | 26.57 | 26.17 | 24.00 |
| 35, 45 | Naive | 27.74 | 27.17 | 26.66 | 26.24 | 26.75 | 25.61 | 24.75 |
| 35, 45 | Refined | **29.14** | **28.33** | **27.67** | **27.19** | **27.69** | **26.61** | **25.88** |

As we can see, the refined policy brings the best performance on all the noise levels including 40. The restoration unit trained for specific noise level (*i.e.* $\sigma = 40$) is only comparable to the one with refined policy on noise level 40. The restoration unit trained on multiple noise levels with naive policy has the worst performance.

These results indicate that the restoration unit has the potential to generalize on unseen degradation levels when trained with good policies. According to Fig. 4, the generalization reflects on the loop times of the restoration unit. It can be observed that the model with steeper slopes have stronger ability to generalize as well as better performances.

According to these results, the restoration unit we used in DURR is trained using the refined policy. More specifically, for image denoising, the noise level and the associated loop times are set to (25, 4), (35, 6), (45, 9), and (55, 12). For JPEG image deblocking, the quality factor (QF) and the associated loop times are set to (20, 6) and (30, 4).

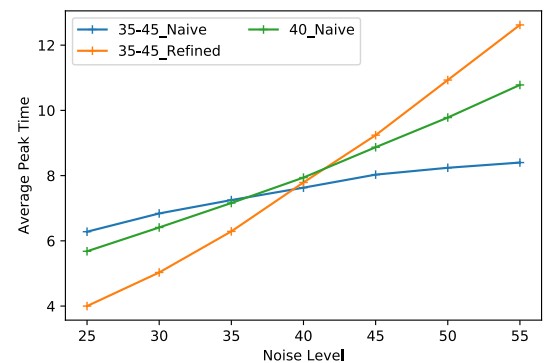

Figure 4: Average peak time on BSD68 with different training strategies.

## 2.2 TRAINING THE POLICY UNIT

We discuss two approaches that can be used as policy unit:

**Handcraft policy:** Previous work (Mrázek & Navara, 2003) has proposed a handcraft policy that selects a terminal time which optimizes the correlation of the signal and noise in the filtered image. This criterion can be used directly as our policy unit, but the independency of signal and noise may

not hold for some restoration tasks such as real image denoising, which has higher noise level in the low-light regions, and JPEG image deblocking, in which artifacts are highly related to the original image. Another potential stopping criterion of the diffusion is no-reference image quality assessment (Mittal et al., 2012), which can provide quality assessment to a processed image without the ground truth image. However, to the best of our knowledge, the performances of these assessments are still far from satisfactory. Because of the limitations of the handcraft policies, we will not include them in our experiments.

**Reinforcement learning based policy:** We start with a discretization of the moving endpoint problem (1) on the dataset $\{(x_i, y_i)|i = 1, 2, \cdots, d\}$, where $\{x_i\}$ are degraded observations of the damage-free images $\{y_i\}$. The discrete moving endpoint control problem is given as follows:

$$
\min_{w, \{N_i\}_{i=1}^d} r(w) + \sum_{i=1}^{d} L(X_{N_i}^i, y_i)
$$
$$
s.t. \ X_n^i = X_{n-1}^i + \Delta t f(X_{n-1}^i, w), n = 1, 2, \cdots, N_i, (i = 1, 2, \cdots, d) \tag{3}
$$
$$
X_0^i = x_i, i = 1, 2, \cdots, d.
$$

Here, $X_n^i = X_{n-1}^i + \Delta t f(X_{n-1}^i, w)$ is the forward Euler approximation of the dynamics $\dot{X} = f(X(t), w)$. The terminal time $\{N_i\}$ is determined by a policy network $P(x, \theta)$, where $x$ is the output of the restoration unit at each iteration and $\theta$ the set of weights. In our experiment, we simply set $r = 0$, *i.e.* doesn't introduce any regularization which might bring further benefit but is beyond this paper's scope of discussion. In other words, the role of the policy network is to stop the iteration of the restoration unit when an ideal image restoration result is achieved. The reward function of the policy unit can be naturally defined by

$$
r(\{X_n^i\}) = \begin{cases} \lambda \left( L(x_{n-1}, y_i) - L(x_n, y_i) \right) & \text{If choose to continue} \\ 0 & \text{Otherwise} \end{cases} \tag{4}
$$

In order to solve the problem (2.2), we need to optimize two networks simultaneously, *i.e.* the restoration unit and the policy unit. The first is an restoration unit which approximates the controlled dynamics and the other is the policy unit to give the optimized terminating conditions. The objective function we use to optimize the policy network can be written as

$$
J = \mathbb{E}_{X \sim \pi_\theta} \sum_n^{N_i} [r(\{X_n^i, w\})], \tag{5}
$$

where $\pi_\theta$ denotes the distribution of the trajectories $X = \{X_n^i, n = 1, \ldots, N_i, i = 1, \ldots, d\}$ under the policy network $P(\cdot, \theta)$. Thus, reinforcement learning techniques can be used here to learn a neural network to work as a policy unit. We utilize Deep Q-learning (Mnih et al., 2015) as our learning strategy and denote this approach simply as **DURR**. However, different learning strategies can be used (e.g. the Policy Gradient).

## 3 EXPERIMENTS

### 3.1 EXPERIMENT SETTINGS

In all denoising experiments, we follow the same settings as in Chen & Pock (2017); Zhang et al. (2017a); Lefkimmiatis (2017). All models are evaluated using the mean PSNR as the quantitative metric on the BSD68 (Martin et al., 2001). The training set and test set of BSD500 (400 images) are used for training. Six gaussian noise levels are evaluated, namely $\sigma = 25, 35, 45, 55, 65$ and $75$. Additive noise are applied to the image on the fly during training and testing. Both the training and evaluation process are done on gray-scale images.

The restoration unit is a simple U-Net (Ronneberger et al., 2015) style fully convolutional neural network. For the training process of the restoration unit, the noise levels of 25, 35, 45 and 55 are

used. Images are cut into $64 \times 64$ patches, and the batch-size is set to 24. The Adam optimizer with the learning rate 1e-3 is adopted and the learning rate is scaled down by a factor of 10 on training plateaux.

The policy unit is composed of two ResUnit and an LSTM cell. For the policy unit training, we utilize the reward function in Eq.4. For training the policy unit, an RMSprop optimizer with learning rate 1e-4 is adopted. We've also tested other network structures, these tests and the detailed network structures of our model are demonstrated in the supplementary materials.

In all JPEG deblocking experiments, we follow the settings as in Zhang et al. (2017a; 2018). All models are evaluated using the mean PSNR as the quantitative metric on the LIVE1 dataset (Sheikh, 2005). Both the training and evaluation processes are done on the Y channel (the luminance channel) of the YCbCr color space. The PIL module of python is applied to generate JPEG-compressed images. The module produces numerically identical images as the commonly used MATLAB JPEG encoder after setting the quantization tables manually. The images with quality factors 20 and 30 are used during training. De-blocking performances are evaluated on four quality factors, namely QF = 10, 20, 30, and 40. All other parameter settings are the same as in the denoising experiments.

## 3.2 Image Denoising

We select DnCNN-B(Zhang et al., 2017a) and UNLNet$_5$ (Lefkimmiatis, 2017) for comparisons since these models are designed for blind image denoising. Moreover, we also compare our model with non-learning-based algorithms BM3D (Dabov et al., 2007) and WNNM (Gu et al., 2014). The noise levels are assumed known for BM3D and WNNM due to their requirements. Comparison results are shown in Table 2.

Despite the fact that the parameters of our model ($1.8 \times 10^5$ for the restoration unit and $1.0 \times 10^5$ for the policy unit) is less than the DnCNN (approximately $7.0 \times 10^5$), one can see that DURR outperforms DnCNN on most of the noise-levels. More interestingly, DURR does not degrade too much when the the noise level goes beyond the level we used during training. The noise level $\sigma = 65, 75$ is not included in the training set of both DnCNN and DURR. DnCNN reports notable drops of PSNR when evaluated on the images with such noise levels, while DURR only reports small drops of PSNR (see the last row of Table 2 and Fig. 6). Note that the reason we do not provide the results of UNLNet$_5$ in Table 2 is because the authors of Lefkimmiatis (2017) has not released their codes yet, and they only reported the noise levels from 15 to 55 in their paper. We also want to emphasize that they trained two networks, one for the low noise level ($5 \leq \sigma \leq 29$) and one for higher noise level ($30 \leq \sigma \leq 55$). The reason is that due to the use of the constraint $||y - x||_2 \leq \epsilon$ by Lefkimmiatis (2017), we should not expect the model generalizes well to the noise levels surpasses the noise level of the training set.

For qualitative comparisons, some restored images of different models on the BSD68 dataset are presented in Fig. 5 and Fig. 6. As can be seen, more details are preserved in DURR than other models. It is worth noting that the noise level of the input image in Fig. 6 is 65, which is unseen by both DnCNN and DURR during training. Nonetheless, DURR achieves a significant gain of nearly 1 dB than DnCNN. Moreover, the texture on the cameo is very well restored by DURR. These results clearly indicate the strong generalization ability of our model.

More interestingly, due to the generalization ability in denoising, DURR is able to handle the problem of real image denoising without additional training. For testing, we test the images obtained from Lebrun et al. (2015). We present the representative results in Fig. 7 and more results are listed in the supplementary materials.

We also train our model for blind color image denoising, please refer to the supplementary materials for more details.

## 3.3 JPEG Image Deblocking

For deep learning based models, we select DnCNN-3 (Zhang et al., 2017a) for comparisons since it is the only known deep model for multiple QFs deblocking. As the AR-CNN (Dong et al., 2015) is a commonly used baseline, we re-train the AR-CNN on a training set with mixed QFs and denote

Table 2: Average PSNR (dB) results for gray image denoising on the BSD68 dataset. Values with * means the corresponding noise level is not present in the training data of the model. The best results are indicated in red.

|  | BM3D | WNNM | DnCNN-B | UNLNet$_5$ | DURR |
|---|---|---|---|---|---|
| $\sigma = 25$ | 28.55 | 28.73 | 29.16 | 28.96 | 29.16 |
| $\sigma = 35$ | 27.07 | 27.28 | 27.66 | 27.50 | 27.72 |
| $\sigma = 45$ | 25.99 | 26.26 | 26.62 | 26.48 | 26.71 |
| $\sigma = 55$ | 25.26 | 25.49 | 25.80 | 25.64 | 25.91 |
| $\sigma = 65$ | 24.69 | 24.51 | 23.40* | - | 25.26* |
| $\sigma = 75$ | 22.63 | 22.71 | 18.73* | - | 24.71* |

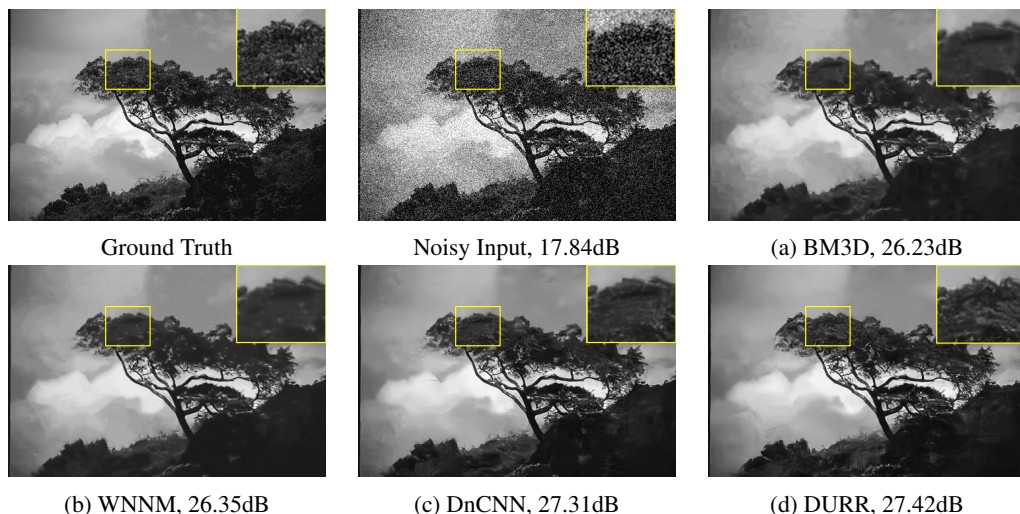

Figure 5: Denoising results of an image from BSD68 with noise level 35.

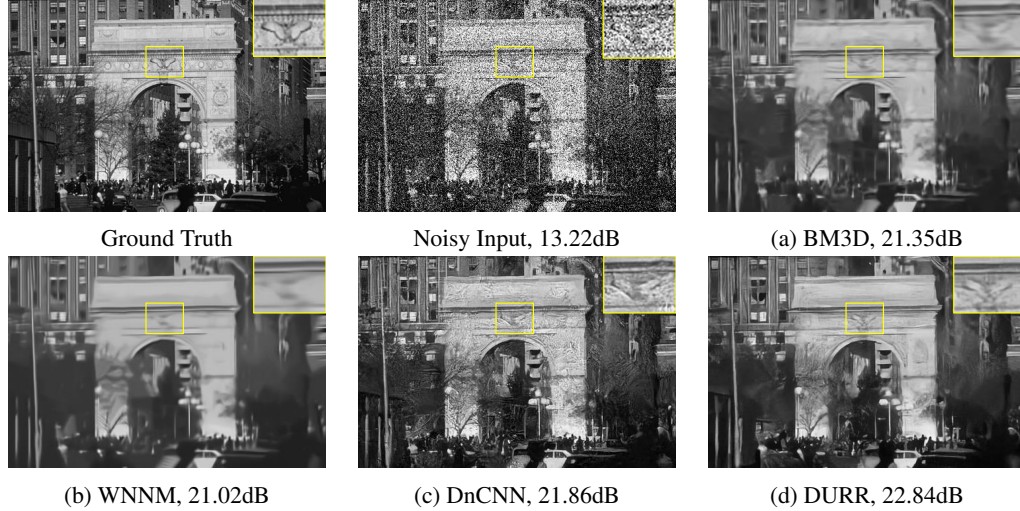

Figure 6: Denoising results of an image from BSD68 with noise level 65 (unseen by both DnCNN and DURR in their training sets).

this model as AR-CNN-B. Original AR-CNN as well as a non-learning-based method SA-DCT (Foi et al., 2007) are also tested. The quality factors are assumed known for these models.

Quantitative results are shown in Table 3. Though the number of parameters of DURR is significantly less than the DnCNN-3, the proposed DURR outperforms DnCNN-3 in most cases. Specifically, considerable gains can be observed for our model on seen QFs, and the performances are comparable on unseen QFs. A representative result on the LIVE1 dataset is presented in Fig. 8. Our model generates the most clean and accurate details. More experiment details are given in the supplementary materials.

Table 3: The average PSNR(dB) on the LIVE1 dataset. Values with * means the corresponding QF is not present in the training data of the model. The best results are indicated in red and the second best results are indicated in blue.

| QF | JPEG | SA-DCT | AR-CNN | AR-CNN-B | DnCNN-3 | DURR |
|----|-------|--------|--------|----------|---------|------|
| 10 | 27.77 | 28.65 | 28.98 | 28.53 | 29.40 | 29.23* |
| 20 | 30.07 | 30.81 | 31.29 | 30.88 | 31.59 | 31.68 |
| 30 | 31.41 | 32.08 | 32.69 | 32.31 | 32.98 | 33.05 |
| 40 | 32.45 | 32.99 | 33.63 | 33.39 | 33.96 | 34.01* |

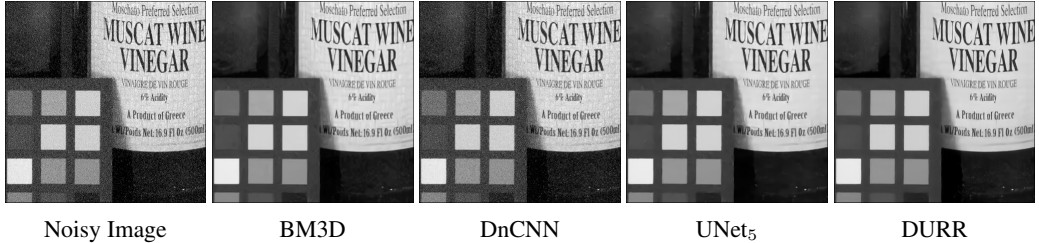

Noisy Image          BM3D          DnCNN          UNet$_5$          DURR

Figure 7: Denoising results on a real image from Lebrun et al. (2015).

## 3.4 OTHER APPLICATIONS

Our model can be easily extended to other applications such as deraining, dehazing and deblurring. In all these applications, there are images corrupted at different levels. Rainfall intensity, haze density and different blur kernels will all effect the image quality.

## 4 CONCLUSIONS

In this paper, we proposed a novel image restoration model based on the moving endpoint control in order to handle varied noise levels using a single model. The problem was solved by jointly optimizing two units: restoration unit and policy unit. The restoration unit used an RNN to realize the dynamics in the control problem. A policy unit was proposed for the policy unit to determine the loop times of the restoration unit for optimal results. Our model achieved the state-of-the-art results in blind image denoising and JPEG deblocking. Moreover, thanks to the flexibility of the given policy, DURR has shown strong abilities of generalization in our experiments.

## ACKNOWLEDGMENTS

Bin Dong is supported in part by Beijing Natural Science Foundation (Z180001).Yiping Lu is supported by the Elite Undergraduate Training Program of the School of Mathematical Sciences at Peking University.

## REFERENCES

Michal Aharon, Michael Elad, and Alfred Bruckstein. $k$-svd: An algorithm for designing overcomplete dictionaries for sparse representation. *IEEE Transactions on signal processing*, 54(11):4311–4322, 2006.

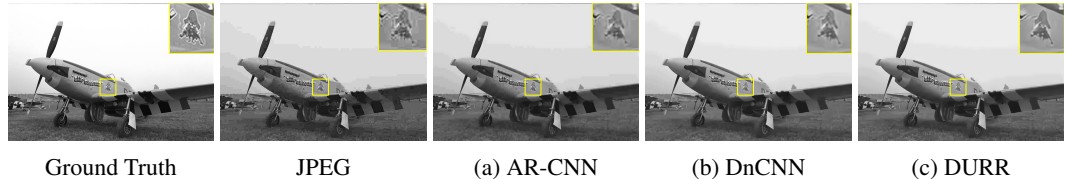

| Ground Truth | JPEG | (a) AR-CNN | (b) DnCNN | (c) DURR |

Figure 8: JPEG deblocking results of an image from the LIVE1 dataset, compressed using QF 10.

K. Bredies, K. Kunisch, and T. Pock. Total Generalized Variation. *SIAM Journal on Imaging Sciences*, 3:492, 2010.

Antoni Buades, Bartomeu Coll, and J-M Morel. A non-local algorithm for image denoising. In *Computer Vision and Pattern Recognition, 2005. CVPR 2005. IEEE Computer Society Conference on*, volume 2, pp. 60–65. IEEE, 2005.

J.F. Cai, S. Osher, and Z. Shen. Split Bregman methods and frame based image restoration. *Multiscale Modeling and Simulation: A SIAM Interdisciplinary Journal*, 8(2):337–369, 2009.

Jian-Feng Cai, Bin Dong, Stanley Osher, and Zuowei Shen. Image restoration: total variation, wavelet frames, and beyond. *Journal of the American Mathematical Society*, 25(4):1033–1089, 2012.

Jian-Feng Cai, Bin Dong, and Zuowei Shen. Image restoration: a wavelet frame based model for piecewise smooth functions and beyond. *Applied and Computational Harmonic Analysis*, 41(1):94–138, 2016.

Francine Catté, Pierre-Louis Lions, Jean-Michel Morel, and Tomeu Coll. Image selective smoothing and edge detection by nonlinear diffusion. *SIAM Journal on Numerical analysis*, 29(1):182–193, 1992.

Bo Chang, Lili Meng, Eldad Haber, Lars Ruthotto, David Begert, and Elliot Holtham. Reversible architectures for arbitrarily deep residual neural networks. *AAAI2018*, 2017.

Qifeng Chen, Jia Xu, and Vladlen Koltun. Fast image processing with fully-convolutional networks. In *Proceedings of the IEEE International Conference on Computer Vision*, pp. 2497–2506, 2017.

Y. Chen and T Pock. Trainable nonlinear reaction diffusion: A flexible framework for fast and effective image restoration. *IEEE Transactions on Pattern Analysis & Machine Intelligence*, 39(6):1256–1272, 2017.

Kostadin Dabov, Alessandro Foi, Vladimir Katkovnik, and Karen Egiazarian. Image denoising by sparse 3-d transform-domain collaborative filtering. *IEEE Transactions on image processing*, 16(8):2080–2095, 2007.

I. Daubechies, G. Teschke, and L. Vese. Iteratively solving linear inverse problems under general convex constraints. *Inverse Problems and Imaging*, 1(1):29, 2007.

Bin Dong, Qingtang Jiang, and Zuowei Shen. Image restoration: Wavelet frame shrinkage, nonlinear evolution pdes, and beyond. *Multiscale Modeling & Simulation*, 15(1):606–660, 2017.

Chao Dong, Yubin Deng, Chen Change Loy, and Xiaoou Tang. Compression artifacts reduction by a deep convolutional network. In *Proceedings of the IEEE International Conference on Computer Vision*, pp. 576–584, 2015.

M. Elad, J.L. Starck, P. Querre, and D.L. Donoho. Simultaneous cartoon and texture image inpainting using morphological component analysis (MCA). *Applied and Computational Harmonic Analysis*, 19(3):340–358, 2005.

Lawrence C Evans. An introduction to mathematical optimal control theory version 0.2. *Tailieu Vn*, 2005.

Cong Fang, Zhenyu Zhao, Pan Zhou, and Zhouchen Lin. Feature learning via partial differential equation with applications to face recognition. *Pattern Recognition*, 69:14–25, 2017.

Alessandro Foi, Vladimir Katkovnik, and Karen Egiazarian. Pointwise shape-adaptive dct for high-quality denoising and deblocking of grayscale and color images. *IEEE Transactions on Image Processing*, 16(5):1395–1411, 2007.

Shuhang Gu, Lei Zhang, Wangmeng Zuo, and Xiangchu Feng. Weighted nuclear norm minimization with application to image denoising. In *Proceedings of the IEEE Conference on Computer Vision and Pattern Recognition*, pp. 2862–2869, 2014.

Kaiming He, Xiangyu Zhang, Shaoqing Ren, and Jian Sun. Deep residual learning for image recognition. In *Proceedings of the IEEE conference on computer vision and pattern recognition*, pp. 770–778, 2016.

Viren Jain and Sebastian Seung. Natural image denoising with convolutional networks. In *Advances in Neural Information Processing Systems*, pp. 769–776, 2009.

H. Ji, C. Liu, Z. Shen, and Y. Xu. Robust video denoising using low rank matrix completion. *IEEE Conference on Computer Vision and Pattern Recognition (CVPR)*, 2010.

Xiangyang Lan, Stefan Roth, Daniel Huttenlocher, and Michael J Black. Efficient belief propagation with learned higher-order markov random fields. In *European conference on computer vision*, pp. 269–282. Springer, 2006.

Marc Lebrun, Miguel Colom, and Jean-Michel Morel. The noise clinic: a blind image denoising algorithm. *Image Processing On Line*, 5:1–54, 2015.

Stamatios Lefkimmiatis. Universal denoising networks: A novel cnn-based network architecture for image denoising. *arXiv preprint arXiv:1711.07807*, 2017.

Qianli Liao and Tomaso Poggio. Bridging the gaps between residual learning, recurrent neural networks and visual cortex. *arXiv preprint*, 2016.

Yiping Lu, Aoxiao Zhong, Quanzheng Li, and Bin Dong. Beyond finite layer neural networks: Bridging deep architectures and numerical differential equations. *Thirty-fifth International Conference on Machine Learning (ICML)*, 2018.

D. Martin, C. Fowlkes, D. Tal, and J. Malik. A database of human segmented natural images and its application to evaluating segmentation algorithms and measuring ecological statistics. In *Proc. 8th Int'l Conf. Computer Vision*, volume 2, pp. 416–423, July 2001.

Anish Mittal, Anush Krishna Moorthy, and Alan Conrad Bovik. No-reference image quality assessment in the spatial domain. *IEEE Transactions on Image Processing*, 21(12):4695–4708, 2012.

Volodymyr Mnih, Koray Kavukcuoglu, David Silver, Andrei A Rusu, Joel Veness, Marc G Bellemare, Alex Graves, Martin Riedmiller, Andreas K Fidjeland, Georg Ostrovski, et al. Human-level control through deep reinforcement learning. *Nature*, 518(7540):529, 2015.

Pavel Mrázek and Mirko Navara. Selection of optimal stopping time for nonlinear diffusion filtering. *International Journal of Computer Vision*, 52(2-3):189–203, 2003.

D. Mumford and J. Shah. Optimal approximations by piecewise smooth functions and associated variational problems. *Communications on pure and applied mathematics*, 42(5):577–685, 1989.

Stanley Osher and Leonid Rudin. Feature-oriented image enhancement using shock filters. *SIAM Journal on Numerical Analysis*, 27(4):919–940, Aug 1990. URL http://www.jstor.org/stable/2157689.

Pietro Perona and Jitendra Malik. Scale-space and edge detection using anisotropic diffusion. *IEEE Transactions on pattern analysis and machine intelligence*, 12(7):629–639, 1990.

Olaf Ronneberger, Philipp Fischer, and Thomas Brox. U-net: Convolutional networks for biomedical image segmentation. In *International Conference on Medical image computing and computer-assisted intervention*, pp. 234–241. Springer, 2015.

Leonid I Rudin, Stanley Osher, and Emad Fatemi. Nonlinear total variation based noise removal algorithms. *Physica D: nonlinear phenomena*, 60(1-4):259–268, 1992.

HR Sheikh. Live image quality assessment database release 2. *http://live.ece.utexas.edu/research/quality*, 2005.

J.L. Starck, M. Elad, and D.L. Donoho. Image decomposition via the combination of sparse representations and a variational approach. *IEEE transactions on image processing*, 14(10):1570–1582, 2005.

Joachim Weickert. *Anisotropic diffusion in image processing*, volume 1. Teubner Stuttgart, 1998.

E Weinan. A proposal on machine learning via dynamical systems. *Communications in Mathematics & Statistics*, 5(1):1–11, 2017.

Kai Zhang, Wangmeng Zuo, Yunjin Chen, Deyu Meng, and Lei Zhang. Beyond a gaussian denoiser: Residual learning of deep cnn for image denoising. *IEEE Transactions on Image Processing*, 26(7):3142–3155, 2017a.

Xiaoshuai Zhang, Wenhan Yang, Yueyu Hu, and Jiaying Liu. Dmcnn: Dual-domain multi-scale convolutional neural network for compression artifacts removal. In *Proceedings of the 25th IEEE International Conference on Image Processing*, 2018.

Xingcheng Zhang, Zhizhong Li, Chen Change Loy, and Dahua Lin. Polynet: A pursuit of structural diversity in very deep networks. In *2017 IEEE Conference on Computer Vision and Pattern Recognition (CVPR)*, pp. 3900–3908. IEEE, 2017b.

