# OpenReview forum: "Dynamically Unfolding Recurrent Restorer: A Moving Endpoint Control Method for Image Restoration"
_ICLR.cc/2019/Conference_

### Official Review · AnonReviewer3 · 2018-10-26
**Simple and effective restoration approach**

**Rating:** 7
**Confidence:** 4

**Review:**

Summary

This paper decomposes the image restoration task in two part: the restoration part handled by a restoration RNN, and the number of steps to apply the RNN is determined using a policy unit.
State of the art results are achieved on blind grey level Gaussian noise denoising on the BSD68 dataset.

The approach is novel to my knowledge, the paper is well written, the results are good and well illustrated.

Questions:
-It would be nice to present results on color images, and on datasets that contains natural noises.
-Lowering the learning rate on plateaus during training is done by hand or is there an automatic criterion to define the plateaus?

Minor:
page 1: extra ")" after ref to Bredies et al 2010
could cite Chen, Zu, Koltun ICCV17 in deep models for restoration
Several "L" have been replaced by "_' e.g. under review at IC_R, R_-based, etc in the whole paper
p.4: rain-> train
greatly influence -> greatly influences
p5: typo performace
make a uniform bib: whole first name or abbr. , no URL, etc.
p6: the weight -> the set of weights
add the specification that the noise is Gaussian
the sentence "the training set and testing set of ..." is used twice, remove one.
p7 Table 1: the perf of DnCNN-B is 29.16 and not 29.15 for sigma 25, right?

---

> ### Author Response · Authors · 2018-11-23
> **Response To Review#3**
>
> We would like to thank Reviewer 3 for their review and helpful suggestions. Our responses inline:
>
> > It would be nice to present results on color images, and on datasets that contains natural noises.
>
> - Of course, we will add results on color images and natural noise datasets in our next revision if time permits. As this requires modifications to the network arch and re-training.
>
> > Lowering the learning rate on plateaus during training is done by hand or is there an automatic criterion to define the plateaus?
>
> - This is done automatically by the pytorch learning rate scheduler (ReduceLROnPlateau), with default arguments.
>
> > p7 Table 1: the perf of DnCNN-B is 29.16 and not 29.15 for sigma 25, right?
>
> - Sorry for the mistake, the perf of DnCNN was from a pytorch re-implementation (code at https://github.com/SaoYan/DnCNN-PyTorch), which gave the result of 29.15. We are referring to the original paper now.
>
> Thank you for your careful reading. We will fix the typos and minor issues in our next revision.

---

### Official Review · AnonReviewer2 · 2018-11-03
**A useful method for blind image restoration problems**

**Rating:** 6
**Confidence:** 5

**Review:**

Summary:

The authors proposes a new image restoration method that becomes particularly useful for blind restoration setting, e.g., the unknown noise variance setting for denoising. They utilized the moving endpoint control methodology, which essentially is applying reinforcement learning to the image restoration, and devised a method that adaptively decides the unfolding steps for given noisy image. The experimental results show encouraging results.

Pros:
In the experimental result, the proposed DURR outperforms DnCNN-B, a current state-of-the-art. Particularly, while DnCNN-B suffers for the noise level that it was not trained for, DURR can still denoise much better. (Table 2) A similar result is obtained for the JPEG deblocking problem as well.

Cons:
- Since the Deep Q-learning is used to train the policy unit, I suspect the training time could be quite long. How does the reward curve look like while training? How stable is the training? Showing such details should make the paper stronger.
- It will be interesting to see more details on the model. For example, what is the mean/std for the number of folds that model applies for BSD68? What is the distribution (histogram?) of the folds for BSD68? Currently, the paper just simply shows the results and seems to hide many details.
- What was the choice for \lambda in Eq. (3),(4)? How do you choose it?
- How does the result look like on other benchmark datasets other than BSD68? It seems like the specific number of looks for each noise level is important for training. Do the choices of (25,4),(35,6),(45,9),(55,12) generalize well to other datasets as well?

Overall, I think the paper should add more details mentioned above to make the paper stronger.

---

> ### Author Response · Authors · 2018-11-23
> **Response To Reviewer#2**
>
> We would like to thank Reviewer 2 for their review and helpful suggestions. Our responses inline:
>
> > Since the Deep Q-learning is used to train the policy unit, I suspect the training time could be quite long. How does the reward curve look like while training? How stable is the training? Showing such details should make the paper stronger.
>
> - The training process is stable and takes only ~ 4 hours on a single Titan Xp GPU for the DQN policy (our main exp). We also tested policy gradient based policies, and the training process is not as stable and needs 2x time to reach similar restoration performances. These details will be included in our next revision, if space permits. However, the details of the employed policy unit are not our main focus (as you can also use handcrafted policies/statistics-based policies as mentioned in Section 2.2). Also, The reward curves look normal and uninformative, as in any other RL tasks.
>
> > It will be interesting to see more details on the model. For example, what is the mean/std for the number of folds that model applies for BSD68? What is the distribution (histogram?) of the folds for BSD68? Currently, the paper just simply shows the results and seems to hide many details.
>
> - This makes a very good point, we've already included part of these details (in Fig. 13). In addition, the variances of the number of unfoldings are quite low for seen noise levels, but larger for those not included in the training process. We’ll improve the figures while introducing new figures to illustrate the behavior of the DURR in our next revision.
>
> > What was the choice for \lambda in Eq. (3),(4)? How do you choose it?
>
> - In our experiements, we simply set \lambda to 0. The regularization term may have potential benefit to the problem and need further studies, though.
>
> > How does the result look like on other benchmark datasets other than BSD68? It seems like the specific number of looks for each noise level is important for training. Do the choices of (25,4), (35,6), (45,9), (55,12) generalize well to other datasets as well?
>
> - Yes, these choices generalize to other datasets. We've tested our performance of the LIVE1 and Set12, and witnessed a similar gain compared to other methods (our own implementation). We only include results on the BSD68 dataset as it is a common practice for denoising papers. Our preliminary results on Set12:
>
> \sigma=25
> BM3D   WNNM   DnCNN-B   DURR (ours)
> 29.97     30.26      30.36           30.41
>
>
> - The choices are only based on empirical results. Using a different training unfolding assignments does not significantly affect the performances though (e.g. using (25,3), (35,5), (45,8), (55,11)).

---

### Official Review · AnonReviewer4 · 2018-11-10
**An improvement to the SoA of the domain, more explanations and analysis welcomed**

**Rating:** 6
**Confidence:** 3

**Review:**

The paper proposes a restoration method based on deep reinforcement learning. It is the idea of trainable unfolding that motivates the use of Reinforcement learning, the restoration unit is a SoA U-Net.

Remarks

* The author seems to make strong assumptions on the nature of the noise and made no attempt to understand the nature of the learning beyond a limited set of qualitative example and PSNR.

* Even if the experimental protocol has been taken from prior work, it would have been appreciated to make it explicit in the paper, especially as ICLR is not a conference of image processing. Indeed, It would have made the paper more self-sufficient.

* Second 2 describing the method is particularly hard to understand and would require more details.

* In the experimental section, the authors claim that "These results indicate that the restoration unit has the potential to generalize on unseen degradation levels when trained with good policies". It would have been important to mention that such generalization capability seems to occur for the given noise type used in the experiments. I didn't see any explicit attempt to variate the shape of the noise to evaluate the generalization capability of the model.

In conclusion, the paper proposes an interesting method of image denoising through state of the art deep learning model and reinforcement learning algorithm. The main difference with the SoA on the domain is the use of a diffusion dynamics. IMHO, the paper would need more analysis and details on the mentioned section.

---

> ### Author Response · Authors · 2018-11-23
> **Response to Reviewer 4**
>
> We would like to thank Reviewer 4 for their review and helpful suggestions. Our responses inline:
>
> > The author seems to make strong assumptions on the nature of the noise and made no attempt to understand the nature of the learning beyond a limited set of qualitative example and PSNR.
>
> - Using gaussian distribution to simulate the noise statistics is a common setting in the image processing papers, due to the central limit theorem. However, we've shown our results for the case of natural image denoising, demonstrating our efficacy for more complicated noise modelings. Anyway, we will include more results on other noise types (e.g. poisson noise) in our next revision.
>
> > Even if the experimental protocol has been taken from prior work, it would have been appreciated to make it explicit in the paper, especially as ICLR is not a conference of image processing. Indeed, It would have made the paper more self-sufficient.
> > Second 2 describing the method is particularly hard to understand and would require more details.
>
> - Sorry for the writing issues. We will refine and include more details for these parts.
>
> > In the experimental section, the authors claim that "These results indicate that the restoration unit has the potential to generalize on unseen degradation levels when trained with good policies". It would have been important to mention that such generalization capability seems to occur for the given noise type used in the experiments. I didn't see any explicit attempt to variate the shape of the noise to evaluate the generalization capability of the model.
>
> - First, by saying "generalize on unseen degradation levels", we mean the generalization power on different degradation levels (e.g. different \sigma for the gaussian noise) but not different noise statistics. Our extensive experiments have shown this kind of generalization for both tasks (gaussian denoise and JPEG deblocking).
>
> - Second, although we does not take the noise statistics into consideration initially when mentioning "generalize", the results on real image denoising clearly indicates our generalization power for unseen noise types: our model trained on gaussian noise also apply to real image noise (Fig. 7, and Section 4.4). And we will add more results on different noise types to further prove our generalization power, as is mentioned above.
>
> We'd also like to defend that our main contribution is introducing the concept of optimal terminal time in the traditional nonlinear diffusion methods into the context of deep learning. Also, we've done analysis on using different restoration units (Appendix 4.1.2). Even using the plain fully convolutional neural network (ARCNN), the quantitive results are not decreasing. This phenomenon indicates that it is the designed dynamically unfolding strategy that brings the performance gain, using whatever kind of network as the restoration unit is not very important.

---

### Public Comment · (anonymous) · 2018-11-06
**Uninformative review from AnonReviewer3**

I've read this paper carefully. After reading the reviews, I don't think AnonReviewer3 has even read the paper, because the comment from AnonReviewer3 is extremely perfunctory and uninformative, comparing to AnonReviewer2 who provides much more useful information about the drawbacks of this paper. Therefore, I strongly suggest AnonReviewer3 lower his/her confindence from 4 to 2.

---

> ### Public Comment · (anonymous) · 2018-11-10
> **Agree with your comment**
>
> I agree with your comment that Reviewer3 does not offer any constructive questions or suggestions. I would expect more high-quality reviews rather than these perfunctory remarks.

---

### Public Comment · (anonymous) · 2018-11-07
**Good motivation, and impressive results**

This paper formulate the image restoration process using dynamically unfolding recurrent CNN, and create links with traditional non-linear diffusion methods. The idea is very natural and intuitive, and shown impressive performance and generalization especially under extreme conditions.A minor question is, does this generalization apply only for denoising? It would be appreciated if you could release more results on different tasks. Thanks.

---

> ### Author Response · Authors · 2018-11-23
> **Response To Anonymous Reader**
>
> Thank you for your kind response. As demonstrated in Section 3.2.2, our generalization power can be witnessed for the task of JPEG deblocking as well. We will add results on other tasks (e.g. de-rain, and deblur), if time permits.

---

### Author Response · Authors · 2018-12-07
**Update of Results on Color Image Gaussian Denoising**

We have done our experiments on color image gaussian denoising, the quantitative results are reported as follows:

\sigma      25       35       45       55        65*      75*
CBM3D     30.71  28.89  27.83  26.97  26.29   25.74
CDnCNN   31.22  29.57  28.40  27.46  26.40  24.47
CDURR      31.25  29.63  28.48  27.57  26.83  26.15

(* The noise level is not presented during training for CDnCNN and DURR)

All the settings are the same as the grayscale experiments, except that the models (CDnCNN and CDURR) are trained on color images. One can see that the CDURR achieves best performance among all, as expected. Qualitative results also show our superior performance compared to other models, on color images and more noise types. These results as well as more details will be included when we have the chance to update the manuscript. Thanks!

---

### Public Comment · (anonymous) · 2019-07-28
**Broken Code Link**

Nice paper. Well motivated, well written, and interesting with good experiments. This solution to the problem will have impact in specific vision problem domains.

However, the code link provided seems to be broken (i.e. https://github.com/BuriedJet/DURR/), Could you fix the problem that would be beneficial to potential readers in this domain?

---

### Meta-Review · Area_Chair1 · 2018-12-13
**novel approach with convincing results**

**Confidence:** 4
**Recommendation:** Accept (Poster)

**Metareview:**

1. Describe the strengths of the paper.  As pointed out by the reviewers and based on your expert opinion.

- The approach is novel
- The experimental results are convincing.

2. Describe the weaknesses of the paper. As pointed out by the reviewers and based on your expert opinion. Be sure to indicate which weaknesses are seen as salient for the decision (i.e., potential critical flaws), as opposed to weaknesses that the authors can likely fix in a revision.

- The authors didn't show results with non-Gaussian noise
- Some details that could help the understanding of the method are missing.

3. Discuss any major points of contention. As raised by the authors or reviewers in the discussion, and how these might have influenced the decision. If the authors provide a rebuttal to a potential reviewer concern, it’s a good idea to acknowledge this and note whether it influenced the final decision or not. This makes sure that author responses are addressed adequately.

No major points of contention.

4. If consensus was reached, say so. Otherwise, explain what the source of reviewer disagreement was and why the decision on the paper aligns with one set of reviewers or another.

The reviewers reached a consensus that the paper should be accepted.